# Investigation of the Roles of the Adenosine A(2A) and Metabotropic Glutamate Receptor Type 5 (mGlu5) Receptors in Prepulse Inhibition and CREB Signaling in a Heritable Rodent Model of Psychosis

**DOI:** 10.3390/cells14030182

**Published:** 2025-01-24

**Authors:** Anthony M. Cuozzo, Loren D. Peeters, Cristal D. Ahmed, Liza J. Wills, Justin T. Gass, Russell W. Brown

**Affiliations:** Department of Biomedical Sciences, James H. Quillen College of Medicine, East Tennessee State University, Johnson City, TN 37614, USA; cuozzoa@etsu.edu (A.M.C.); johnsonc15@etsu.edu (C.D.A.);

**Keywords:** mGlu5, A(2A), dopamine, CREB, prepulse inhibition, psychosis

## Abstract

The metabotropic glutamate receptor type 5 (mGlu5) and adenosine A(2A) receptor form a mutually inhibitory heteromer with the dopamine D2 receptor, where the activation of either mGlu5 or A(2A) leads to reduced D2 signaling. This study investigated whether a mGlu5-positive allosteric modulator (PAM) or an A(2A) agonist treatment could mitigate sensorimotor gating deficits and alter cyclic AMP response element-binding protein (CREB) levels in a rodent neonatal quinpirole (NQ) model of psychosis. F0 Sprague–Dawley rats were treated with neonatal saline or quinpirole (1 mg/kg) from postnatal day 1 to 21 and bred to produce an F1 generation. F1 offspring underwent prepulse inhibition (PPI) testing from postnatal day 44 to 48 to assess sensorimotor gating. The rats were treated with mGlu5 PAM 3-Cyano-N-(1,3-diphenyl-1H-pyrazol-5-yl) benzamide (CDPPB) or A(2A) agonist CGS21680. Rats with at least one NQ-treated parent showed PPI deficits, which were alleviated by both CDPPB and CGS21680. Sex differences were noted across groups, with CGS21680 showing greater efficacy than CDPPB. Additionally, CREB levels were elevated in the nucleus accumbens (NAc), and both CDPPB and CGS21680 reduced CREB expression to control levels. These findings suggest that targeting the adenosinergic and glutamatergic systems alleviates sensorimotor gating deficits and abnormal CREB signaling, both of which are associated with psychosis.

## 1. Introduction

Adenosine is a nucleoside and neuromodulator that plays a critical role in proper brain function, contributing to sleep regulation, neuroprotection, and mediation of the effects of drug use [1]. Adenosine regulates the release of neurotransmitters at the synapse level via A(1) and A(2A) receptors. A(1) receptors are largely enriched in glutamatergic terminals and exert an inhibitory effect [2], whereas A(2A) receptors are facilitatory and responsible for the integration of dopaminergic and glutamatergic neurotransmission [3]. Thus, the A(2A) receptor presents itself as a possible therapeutic target for the treatment of psychiatric disorders. Previous work in our laboratory has shown that administration of the A(2A) agonist CGS21680 alleviates sensorimotor gating deficits in rats given neonatal quinpirole (NQ) [4], which results in increased dopamine D2 receptor sensitivity throughout the animal’s lifetime. Aberrant sensorimotor gating is a biomarker of schizophrenia (SZ) that was established via prepulse inhibition (PPI) testing in a clinical population [5].

The most widely accepted and implemented theory of SZ pathology is based on DA hyperactivity within the mesolimbic pathway of the brain [6]. This idea is supported by the prescription of D_2_ antagonists as antipsychotics, as well as DA-enhancing drugs inducing delusions and hallucinations that are associated with SZ [7]. However, it is likely that other neurotransmitters and neuromodulators contribute to the pathophysiology of SZ. Adenosine is a modulator of glutamate and DA signaling, and diminished function of the adenosine system can produce a hyperdopaminergic state [8]. A(2A) deficiency in the striatum has been previously linked to an enhanced response to amphetamines, and A(2A) deficiency in the hippocampus produces N-methyl-D-aspartate receptor (NMDAR) hypoactivity that is associated with learning deficits [9].

Glutamate is the most abundant excitatory transmitter in the brain and is heavily involved in mechanisms of neuroplasticity and synaptic reorganization [10]. Metabotropic glutamate receptor type 5 (mGlu5) is a modulatory receptor that plays a role in cognition, reward processing, and synaptic plasticity [11]. The mGlu5 receptor modulates NMDA receptor-mediated signaling and has been implicated in the pathology of SZ [12]. Previous work in our laboratory has demonstrated that the administration of mGlu5-positive allosteric modulator (PAM) CDPPB attenuates sensorimotor gating deficits in NQ-treated rats and restores striatal D_2_ signaling to control levels [13]. In summary, the mGlu5 receptor has been identified as a significant therapeutic target for symptom treatment in psychiatric disorders.

The glutamatergic hypothesis of SZ posits that NMDAR hypofunction in the prefrontal cortex and hippocampus contributes to observed symptomology [14]. The mGlu5 receptor, in particular, has been implicated via genome-wide association studies (GWASs), revealing reduced availability and function of the receptor [15]. Several models of psychosis are induced via NMDAR hypofunction through the administration of ketamine, MK-801, or phencyclidine (PCP). MK-801 has been previously shown to impair cognition and spatial memory in rodents [16]. These findings overwhelmingly suggest that the glutamatergic system is disrupted in SZ; thus, it is a necessary therapeutic target for treating the full range of symptoms and insults in the disorder.

The current standard for SZ treatment is founded in D_2_ antagonism, but these compounds produce significant side effects that include motor impairment, metabolic dysregulation, and tardive dyskinesia [17]. Attenuating hypersensitive D2 signaling while reducing the occurrence of adverse effects is imperative to improving treatment outcomes in this population. The A(2A)-D_2_-mGlu5 heteromer is abundant in the striatum, and synergistic actions between the A(2A) and mGlu5 subunits have previously been suggested [18]. Stimulation of either of these subunits results in an overall reduction in DA signaling, as the heteromer is mutually inhibitory [19]. In the present study, we analyzed both CGS21680, an A(2A) agonist, and CDPPB, a PAM, towards mGlu5. We hypothesize that these compounds work to restore hypofunction within the adenosinergic and glutamatergic systems while also producing a net reduction in D_2_ signaling.

In order to model the enhanced hyperactivity of the D_2_ system in SZ, our laboratory has developed a heritable rodent model of psychosis that is induced by neonatal treatment with quinpirole, a D_2_-like receptor agonist [20]. Recently, our laboratory established the heritability of this NQ model by breeding pairs of NQ- or NS-treated rats. The resulting offspring (F1 generation) is untreated and, remarkably, demonstrates enhanced dopamine D2 signaling and presents with the same phenotypical sensorimotor gating deficit as F0 founders [21]. We have previously shown several consistencies between the NQ model and clinical data in SZ [22,23]. Interestingly, we have also demonstrated an enhanced drug reward sensitivity and behavioral response to nicotine [22], which is based on overwhelming epidemiological data demonstrating the vulnerability of this population towards cigarette smoking [24]. Additionally, F1 rats that are the offspring of two NQ-treated founders demonstrate a 26-fold enrichment of the nicotine addiction Kyoto Encyclopedia of Genes and Genomes (KEGG) pathway in the infralimbic cortex (IfL) compared to controls [25].

PPI of the acoustic startle response is an operational measure of sensorimotor gating that reflects the ability of an individual to filter out unnecessary environmental stimuli [26]. Systemic administration of D2-like agonist quinpirole has been previously shown to induce disruptions of PPI [27]. Deficits in PPI in psychosis have been observed across numerous studies, and the administration of second-generation antipsychotics has been shown to normalize these deficits [28]. PPI is highly translatable between animals and humans and possesses significant cross-species measures that can enhance understanding of the pathophysiology of SZ and related psychoses [29]. Thus, we evaluated the efficacy of CGS 21680 and CDPPB to reduce sensorimotor gating deficits in a heritable model of psychosis.

The transcription factor CREB is heavily involved in gene expression that affects downstream neural circuit function and synaptic plasticity [30]. Within psychosis, deficits in CREB have been previously observed in the dorsolateral prefrontal cortex and cingulate gyrus [31]. Changes in CREB within the NAc have been linked to alterations in gating between emotional stimuli and behavioral responses [32]. In previous work in the F1 generation model, we reported an increase in the cortisol synthesis pathway via RNAseq analysis of the dorsal striatum, which has an abundance of D2-like receptors [21]. This finding is consistent with an enhanced stress response, which plays an important role in the manifestation of psychotic symptoms [33]. The association of CREB with emotional reactivity within the context of PPI can greatly affect the ability to filter out irrelevant sensory stimuli [34]. This enhanced emotional reactivity-induced deficit is hypothesized to be involved in the PPI deficits observed within the NQ model. Given this, in the present study, we analyzed protein levels of CREB via ELISA in the NAc to establish changes in CREB within our model and assess the effects of CDPPB and CGS21680 on this emotional reactivity marker.

## 2. Materials and Methods

### 2.1. Experiment 1: Subjects

For this experiment, a total of 185 Sprague–Dawley animals were used as subjects, with 100 subjects that were males and a total of 85 that were females. These animals were a product of a total of 22 breeder pairs in which male/female rats were neonatally treated with saline (NS) or quinpirole HCl (NQ; 1 mg/kg). All breeders and their offspring were housed in the AALAC-accredited animal colony at the East Tennessee State University Quillen College of Medicine. A total of 1–2 males/females were used per litter to control for within-litter variance All animals were fed regular chow and tap water ad libitum and were kept on a 12/12 light/dark cycle. All animals were tested during the light portion of their daily cycle. All procedures reviewed and approved by the University Committee on Animal Care at East Tennessee State University, which conforms to the NIH Guide on the Care and Use of Animals. Across all behavioral analyses, females and males are presented in separate figures. This was done, in part, because there were significant sex differences in many analyses but also based on the number of groups analyzed.

### 2.2. Experiment 1: Drugs

The drugs used in experiment 1 were Quinpirole HCl (Product# Q102) at a dose of 1 mg/kg and adenosine A(2A) agonist CGS 21680 (Product #199137) at doses of 0.03 mg/kg and 0.09 mg/kg. Both drugs were ordered from Sigma-Aldrich, Inc. (St. Louis, MO, USA). Quinpirole was mixed in 0.9% saline. The dose of quinpirole was chosen based on past work, which has shown a dose of 1 mg/kg i.p. administered from postnatal day (P) 1 to 21 results in an increase of dopamine D2 receptor sensitivity throughout the animal’s lifetime [35]. CGS 21680 (Product # C141) was mixed in a 1:1 mixture of 2% dimethyl sulfur oxide (DMSO) and 0.9% NaCl. The doses of CGS 21680 were based on past work that has shown these doses to be effective in alleviating PPI deficits in F0-generation NQ-treated rats [4]. In experiment 1, both drugs were given through an intraperitoneal (i.p.) route of administration.

### 2.3. Experiments 1 and 2: Research Design

F0-generation breeders, which were the offspring of males and females ordered from Inotiv, Inc. (Indianapolis, IN, USA), produced the F1-generation offspring, which were used as subjects. The day of birth for the F0-generation rats was counted as postnatal day (P)0. These animals were neonatally i.p. administered saline or quinpirole (1 mg/kg) once daily from P1 to 21, and neonatal treatments followed our past work [21]. These animals were raised to P60 with occasional handling by experimenters. Breeding commenced on approximately P60, with one male and one female per cage. The F1-generation offspring of F0-generation breeders were used as subjects. F1-generation animals were untreated. There were four crosses used: two breeders that were both neonatally treated with saline (MSxFS), two breeders that were both neonatally treated with quinpirole (MQxFQ), a male breeder that was neonatally treated with quinpirole bred with a female neonatally treated with saline (MQxFS), and a male breeder that was neonatally treated saline bred with a female that was neonatally treated with quinpirole (MSxFQ).

Only F1-generation animals that were the offspring of at least one NQ-treated founder were given CGS 21680 or CDPPB. The rationale for this is that the focus of this study is to analyze whether these drugs are effective in alleviating PPI deficits in animals expressing increased DAD2 receptor sensitivity. The focus was not on whether these drugs had any effect on sensorimotor gating in controls. Furthermore, adding this factor would result in a four-factor study, which could result in a statistically significant four-way interaction, which is an uninterpretable statistical effect.

Finally, we did not systematically analyze the estrous cycle stage in females because the number of animals would have to be dramatically increased 3- to 4-fold to analyze the impact of the stage of the estrous cycle on sensorimotor gating. Adding this factor would increase the number of factors and complicate the research design.

### 2.4. Experiments 1 and 2: Sensorimotor Gating Apparatus

The Startle Monitor II apparatus and software (Kinder Scientific, Poway, CA, USA) were used for all sensorimotor gating testing across experiments 1 and 2. Within the apparatus, subjects were placed into an enclosed dome (height = 8 cm) that was attached to a stainless-steel platform (11 cm wide × 15 cm long). This platform was mounted in a sound-attenuating chamber (28 cm tall × 30 cm wide × 36 cm depth). Subject responses were recorded in Newtons (N) within a 250 millisecond (ms) timeframe following presentation of a stimulus. The apparatus was calibrated during testing based on the weight of the animal in the enclosure. Each subject was tested within the same apparatus across all five days of behavioral analysis.

### 2.5. Experiments 1 and 2: Prepulse Inhibition (PPI) Methods

PPI behavioral methodology is based on previous work completed by our laboratory [36]. Rats began testing on P44 and completed testing on P48 for a total of 5 days. In experiment 1, subjects were i.p. administered saline and 0.03 mg/kg or 0.09 mg/kg CGS21680 10 min prior to placement in the PPI apparatus. In experiment 2, subjects were subcutaneously (s.c.) administered saline and 10 mg/kg or 30 mg/kg CDPPB 20 min prior to testing. In each daily test, the first 5 min consisted of a habituation period in which white noise (70 dB) was the only stimulus. All animals were then administered 60 total trials consisting of 20 pulse, 30 prepulse, and 10 no-stimulus trials. The pulse trial consisted of a 120 dB startle pulse administered alone to produce a startle response. The prepulse trial was an auditory stimulus 3, 6, or 12 dB above the 70 dB white noise (73, 67, and 82 dB, respectively), followed by the 120 dB startle pulse 250 ms later. The stimulus trial consisted of presentation of stimulus. The startle response of the subject was measured in Newtons (N) via software provided by Kinder Scientific and averaged over the five test days for all trials.

### 2.6. Experiments 1 and 2: Cyclic AMP Response Element Binding Protein (CREB) ELISA

All animals were given one more injection of all drug treatments to be consistent with past work [4]. Approximately 24 h following this final injection, on P50, rats were live-decapitated, and brain tissue was harvested, flash-frozen in ice-cold isopentane, and immediately stored at −80 °C. The NAc was dissected using tissue punches that were placed into pre-weighed 1.5 mL Eppendorf tubes and stored for future analysis. Tissue was homogenized using a Fisher Scientific (Atlanta, GA, USA) Dismembrator 5000 at an amplitude of 50% for 3 s per tissue punch. Prior to homogenization, RIPA buffer with protease and phosphatase inhibitors (all from Sigma-Aldrich: phenylmethylsulfonyl fluoride and product numbers P5726, P0044, and P8340) were added to each sample tube. Following homogenization, the samples were centrifuged twice at 4 °C. CREB was analyzed via an ELISA kit from MyBioSource.com (San Diego, CA, USA; Catalog No.: MBS762390), and the protocol was closely followed. In brief, the provided standard was diluted in a serial dilution from 10 ng/mL to 0.156 ng/mL and run in duplicate. Then, 100 µL per well of homogenized sample was added and run in duplicate. The plate was incubated for 90 min at 37 °C, then washed twice. Subsequently, 100 µL of the biotin-labeled antibody working solution was added to each well (100 µL) and incubated for 60 min at 37 °C. The plate was washed three times, and 100 µL of the HRP–streptavidin conjugate working solution was added to each well and incubated for 60 min at 37 °C. The plate was washed five times, after which 90 µL of TMB substrate solution was added and incubated for 20 min. After this incubation, 50 µL of stop solution was added, and the plate was immediately read at 450 nm on a BioTek elx800 Plate Reader (BioTek, Winooski, VT, USA). The resulting data were calculated according to the given standard curve and converted to pg of CREB protein/mg of tissue.

### 2.7. Experiment 2: Subjects

For this experiment, a total of 186 Sprague–Dawley animals were used as subjects, with 91 subjects that were males a total of 95 that were females. These animals were a product of a total of 22 breeder pairs in which male/female rats were neonatally treated with saline (NS) or quinpirole HCl (NQ; 1 mg/kg). Notably, animals in the MSxFS group are the same subjects that were used in experiment 1. Identically to experiment 1, all breeders and their offspring were housed in the AALAC-accredited animal colony at East Tennessee State University Quillen College of Medicine with a 12/12 light/dark cycle. A total of 1–2 males/females were used per litter to control for within-litter variance, had ad libitum access to food and water, and were tested during the light portion of their daily cycle.

### 2.8. Experiment 2: Drugs

In experiment 2, quinpirole HCl was used again, in addition to CDPPB. CDPPB was ordered from EAG Laboratories (San Diego, CA, USA). CDPPB was dissolved in 10% Tween-80 from Sigma-Aldrich, Inc., and mixed with 0.9% NaCl at doses of 10 mg/kg and 30 mg/kg. Quinpirole was given via i.p. injection in experiment 2, with CDPPB administered s.c. The doses of CDPPB were chosen based on past works that have shown that these doses are behaviorally effective in rats [37], as well as in F0-generation rats given NQ [13].

### 2.9. Experiments 1 and 2: Statistical Analysis

For statistical analysis of PPI, a three-way ANOVA was used as the primary statistic, with sex, group, and decibel of prepulse used as the repeated measures. Group was a combination of the founder pair (MQxFQ, MQxFS, MSxFQ, or MSxFS) and the drug that was given (saline, CGS 21680, or CDPPB). The rationale for analysis in this manner was two-fold. First, CGS 21680 or CDPPB was not administered to the MSxFS control group, simply because the study was not focused on analyzing the effects of either of these two drugs alone on sensorimotor gating in controls. Rather, the study was focused on utilizing these drugs to alleviate sensorimotor gating deficits in F1-generation offspring of NQ- or NS-treated founders. Secondly, including drug treatment as a fourth factor in the ANOVA would significantly complicate the statistical analysis and interpretation. For analysis of the startle response, only a two-way ANOVA was used; because there was not a repeated measure, we averaged the startle over the days of testing. For all post hoc analyses, the Bonferroni test was used to analyze any group differences (*p* = 0.05). Across both experiments, there was an N of 7–12 males/females per group that were behaviorally tested. For CREB analyses, there was an N of 6–7 males/females per group.

## 3. Results

### 3.1. CGS21680–MQxFQ PPI and Acoustic Startle

Prepulse inhibition (%) is presented as a function of the decibel of prepulse and group in Figure 1A (females) and Figure 1C (males). The number of animals per group for females was MSxFS-S = 9, MQxFQ-S = 10, and 6 each for MQxFQ-0.03 and MQxFQ-0.09. For males, the number of animals was MSxFS-S = 10, MQxFQ-S = 11, MQxFQ-0.03 = 6, and MQxFQ-0.09 = 8. A three-way ANOVA revealed significant main effects of sex (F(1,58) = 5.5, *p* < 0.022; group F(3,58) = 46.22, *p* < 0.001) and group × dB of prepulse interaction (F(6,116) = 2.45, *p* < 0.029). Females and males were analyzed separately based on the main effect of sex. Overall, females demonstrated significantly improved performance as compared to males. In the separate analysis of females, there was a significant main effect of group (F(3,27) = 23.41, *p* < 0.001) and a significant two-way group × decibel of prepulse interaction (F(6,54) = 3.34, *p* < 0.007). In the separate analysis of males, a two-way ANOVA revealed a significant main effect of group (F(3,31) = 24.42, *p* < 0.001). For both females and males, at all dB levels of prepulse, MSxFS, MQxFQ-0.03, and MQxFQ-0.09 groups all performed equally and were all significantly improved over the MQxFQ group given vehicle. In other words, both doses of the A(2A) agonist improved performance in MQxFQ rats. Acoustic startle is presented in Figure 1B (females) and Figure 1D (males). No effects on acoustic startle were revealed.

### 3.2. CGS21680–MQxFS PPI and Acoustic Startle

Prepulse inhibition (%) is presented as a function of decibel of prepulse and group in Figure 2A (females) and Figure 2C (males). The number of animals per group for females was MSxFS-S = 9, MQxFS-S = 10, MQxFS-0.03 = 8, and MQxFS-0.09 = 7. For males, the number of animals was MSxFS-S = 10, MQxFS = S = 12, and 6 in MQxFS-0.03 and MQxFS-0.09. A three-way ANOVA revealed a significant main effect of drug (F(3,60) = 17.9, *p* < 0.001), a significant two-way interaction of drug ×dB of prepulse (F(6,120) = 2.64, *p* < 0.019), and a significant three-way interaction of sex × drug × dB of prepulse (F(6,120) = 2.50, *p* < 0.026). We then analyzed the two sexes separately. In females, a two-way ANOVA revealed a significant main effect of group (F(3,30) = 4.95, *p* < 0.007) and a significant two-way interaction of group × decibel of prepulse (F(6,60) = 3.25, *p* < 0.008). In females, at 73 and 76 dB, no difference between MQxFS-0.03 and vehicle were observed, but MQxFS.09 and MSxFS-S were greater than MQxFS-S. No group differences were observed in females at 82 dB. In males, a two-way ANOVA revealed a significant main effect of group (F(3,30) = 16.82, *p* < 0.001). In males, at all three dB levels, the MSxFS, MQxFS-0.03, and MQxFS-0.09 groups all performed equally and were all significantly improved over the MQxFS group given vehicle. In MQxFS, the 0.03 mg/kg dose of CGS 21680 was not effective in females at the lower decibel levels, and deficits in female MQxFS did not demonstrate as severe of PPI deficits as MQxFQ.

Acoustic startle is presented in Figure 2B (females) and Figure 2D (males). A two-way ANOVA revealed a significant main effect of group (F(3,60) = 4.39, *p* < 0.008). The MSxFS group was significantly higher than the MQxFS-0.03 group. Thus, the lower dose of CGS 21680 significantly decreased acoustic startle in the MQxFS group.

### 3.3. CGS21680–MSxFQ PPI and Acoustic Startle

Prepulse Inhibition (%) is presented as a function of decibel of prepulse and group in Figure 3A (females) and Figure 3C (males). The number of animals per group for females was nine in groups MSxFS-S and MSxFQ-S and six in groups MSxFQ-0.03 and MSxFQ-0.09. For males, the number of animals was MSxFS-S = 10, MSxFQ = S-S = 9, and 6 each in MSxFQ-0.03 and MSxFQ-0.09. A three-way repeated-measures ANOVA revealed a significant main effect of group (F(3,51) = 12.78, *p* < 0.001) and significant two-way interactions of sex × decibel of prepulse (F(2,102) = 3.21, *p* < 0.045) and group × decibel of prepulse (F(6,102) = 4.30, *p* < 0.002). Based on the significant two-way interaction of sex × decibel of prepulse, females and males were analyzed separately. In females, a two-way repeated-measures ANOVA revealed a significant main effect of group (F(3,24) = 3.83, *p* < 0.023). In females, at 73 dB and 76 dB, the MSxFS was greater than MSxFQ-S; however, there were no differences between groups given either 0.03 or 0.09 mg/kg CGS 21680 and MSxFS-S or MSxFQ-S at this level of prepulse. At 82 dB, there was no significant group difference. In males, a two-way repeated-measures ANOVA revealed a significant main effect of group (F(3,27) = 14.90, *p* < 0.001) and a significant interaction of group × decibel of prepulse (F(6,54) = 5.08, *p* < 0.001), and MSxFQ-S was significantly below all other male groups at all three decibels of prepulse. Therefore, CGS 21680 was effective in alleviating PPI deficits in MSxFQ males at all decibels of prepulse.

Acoustic startle is presented in Figure 3B (females) and Figure 3D (males). There were no significant effects on startle.

### 3.4. CDPPB–MQxFQ PPI and Acoustic Startle

Prepulse inhibition (%) is presented as a function of decibel of prepulse and group in Figure 4A (females) and Figure 4C (males). The number of animals per group for females was MSxFS-S = 9, MQxFQ-S = 10, and 6 in each of MQxFQ-10 and MQxFQ-30. For males, the number of animals was MSxFS-S = 10, MQxFQ-S = 11, and 6 each in MQxFQ-10 and MQxFQ-30. A three-way repeated-measures ANOVA revealed significant main effects of sex (F(1,56) = 4.7, *p* < 0.034) and group (F(3,56) = 25.74, *p* < 0.001), as well as a significant two-way interaction of group × dB of prepulse F((6,112) = 2.26, *p* < 0.042). Based on the significant main effects of sex, females and males were analyzed separately. In females, a two-way repeated-measures ANOVA revealed a significant main effect of group (F(3,28) = 9.61, *p* < 0.001) and a significant two-way interaction of group × decibel of prepulse (F(6,56) = 2.61, *p* < 0.027). At 73 and 76 dB, MSxFS was equivalent to MQxFQ-10 and MQxFQ-30 and demonstrated improved PPI compared to MQxFQ-S. However, the MQxFQ-10 group was improved compared to MQxFQ-S, but MQxFQ-S was equivalent to MQxFQ-30. At 82 dB, groups MSxFS, MQxFQ-10, and MQxFQ-30 were all significantly greater than MQxFQ-S. It appears the lower dose of 10 mg/kg CDPPB was more effective in females, but the 30 mg/kg dose did not improve PPI until the decibel of prepulse reached 82 dB. In males, a two-way repeated-measures ANOVA revealed a significant main effect of group (F(3,28) = 20.71, *p* < 0.001). At all three levels of prepulse, MSxFS-S and MQxFQ-30 were significantly improved compared to MQxFQ-S, but MQxFQ-S did not significantly differ from MQxFQ-10. Therefore, the 30 mg/kg CDPPB but not the 10 mg/kg dose was effective in alleviating PPI deficits in males.

Acoustic startle is presented in Figure 4B (females) and Figure 4D (males). There were no significant effects on startle.

### 3.5. CDPPB–MQxFS PPI and Acoustic Startle

Prepulse inhibition (%) is presented as a function of decibel of prepulse and group in Figure 5A (females) and Figure 5C (males). The number of animals per group for females was MSxFS-S = 9, MQxFS-S = 10, and 6 each in MQxFS-10 and MQxFS-30. For males, the number of animals was MSxFS-S = 10, MQxFS-S = 12, and 6 each in MQxFS-10 and MQxFS-30. A three-way repeated-measures ANOVA revealed a significant main effect of group (F(3,50) = 23.50, *p* < 0.001) and significant two-way interactions of sex × decibel of prepulse (F(2,100) = 3.60, *p* < 0.031) and group × decibel of prepulse (F(6,100) = 3.43, *p* < 0.004). In females, a two-way repeated-measures ANOVA revealed a significant main effect of group (F(3,25) = 6.63, *p* < 0.001) and a significant two-way interaction of group x decibel of prepulse (F(6,50) = 3.89, *p* < 0.003). In females, a two-way repeated-measures ANOVA revealed a significant main effect of group (F(3,25) = 6.63, *p* < 0.001) and a significant two-way interaction of group x decibel of prepulse (F(6,50) = 3.90, *p* < 0.003). Bonferroni post hoc analyses were conducted, and at 73 and 76 decibel levels of prepulse, MSxFS-S and MQxFS-30 were equivalent and both significantly greater than MQxFS-S and MQxFS-10. At 82 dB prepulse, MSxFS was significantly greater than MQxFS-10 but not different from the MQxFS-30 or MQxFS-S groups. For males, a two-way repeated-measures ANOVA revealed a significant main effect of group (F(3,25) = 26.30, *p* < 0.001). Across all decibel levels of prepulse, MSxFS-S was equivalent to MQxFS-30, and both were significantly greater than the MQxFS-10 and MQxFS-S groups.

Acoustic startle is presented in Figure 5B (females) and Figure 5D (males). Interestingly, with respect to startle response, a two-way ANOVA revealed both significant main effects of sex (F(1,59) = 3.15, *p* < 0.047) and group (F(3,59) = 3.15, *p* < 0.033). Males demonstrated a significantly higher startle response than females, and the MQxFS-10 males demonstrated a significantly higher startle response than the MQxFS-30 group.

### 3.6. CDPPB–MSxFQ PPI and Acoustic Startle

Prepulse inhibition (%) is presented as a function of decibel of prepulse and group in Figure 6A (females) and Figure 6C (males). The number of animals per group for females was nine in groups MSxFS-S and MSxFQ-S and eight in groups MSxFQ-10 and MSxFQ-30. For males, the number of animals was MSxFS-S = 10, MSxFQ-S = 9, and 6 each in MSxFQ-10 and MSxFQ-30. A three-way repeated-measures ANOVA revealed significant main effects of sex (F(1,57) = 6.8, *p* < 0.012) and group (F(3,57) = 10.81, *p* < 0.001) and an interaction of group × decibel of prepulse (F(6,114) = 3.02, *p* < 0.009). For females, a two-way ANOVA revealed a significant main effect of group (F(3,33) = 5.7, *p* < 0.003). Overall, MSxFS-S was equivalent to both the MSxFQ-10 and MSxFQ-30 groups, and MSxFS was significantly greater than the MSxFQ-S group. However, the MSxFQ-S group also did not significantly differ from the MSxFQ-10 or MSxFQ-30 groups. Since the group × decibel of prepulse interaction was not statistically significant, this effect did not affect overall performance across all three decibels of prepulse. In males, a two-way ANOVA revealed a significant main effect of group (F(3,24) = 8.04, *p* < 0.001) and a significant interaction of group × decibel of prepulse (F(6,48) = 2.44, *p* < 0.039). At 73 and 76 dB, MSxFS-S was equivalent to MSxFQ-30, and significantly greater than MSxFQ-10 and MSxFQ-S. At 82 dB, MSxFS-S was significantly greater than MSxFQ-10, but there were no other group differences, demonstrating that MSxFQ-S males did not demonstrate a significant deficit at 82 dB. Males in the MSxFQ group demonstrated a deficit when given the lower dose of CDPPB, which was a surprising result. However, the 30 mg/kg dose was effective. Acoustic startle is presented in Figure 6B (females) Figure 6D (males). There were no significant effects on startle.

### 3.7. A(2A) Agonist: CGS21680–CREB

#### 3.7.1. Numbers of Animals for CREB Analyses

Across all CREB ELISA analyses, there were N values of four males and four females per group. The reason for the reduction in numbers compared to behavior was that all ELISAs were performed in 96-well plates, and there were not enough wells to represent all animals behaviorally tested in this study.

#### 3.7.2. MQxFQ Group

CREB is presented as a function of group in Figure 7A. A two-way ANOVA revealed no significant main effect of sex. Therefore, we dropped sex as a factor, and a one-way ANOVA revealed a robust significant main effect of group (F(3,35) = 22.42, *p* < 0.001). The MQxFQ-S group was significantly higher than all other groups, demonstrating that both doses of CGS 21680 were effective in reducing the significant increase in CREB in the NAc.

#### 3.7.3. MQxFS Group

CREB is presented as a function of group in Figure 7B. A two-way ANOVA revealed no significant main effect of sex. Therefore, we dropped sex as a factor, and a one-way ANOVA revealed a significant main effect of group (F(3,37) = 16.97, *p* < 0.001). The MQxFS-S group was significantly higher than all other groups, and there were no other significant differences between groups. As with the MQxFQ group, both doses of CGS 21680 were effective in reducing CREB in the NAc.

#### 3.7.4. MSxFQ Group

CREB is presented as a function of group in Figure 7C. A two-way ANOVA revealed no significant main effect of sex. Therefore, we dropped sex as a factor, and a one-way ANOVA revealed a significant main effect of group (F(3,33) = 5.09, *p* < 0.011). The MSxFQ-S group was significantly higher for CREB in the NAc than MSxFS-S, but there were no other group differences.

### 3.8. mGlu5 PAM: CDPPB–CREB

#### 3.8.1. MQxFQ Group

CREB is presented as a function of males in Figure 8A and females in Figure 8B. A two-way ANOVA revealed both a significant main effect of sex (F(1,39) = 5.06, *p* < 0.032) and a main effect of group (F(3,39) = 16.19, *p* < 0.001). Overall, females demonstrated a significantly higher level of CREB in the NAc than males, likely owing the lack of efficacy of CDPPB. Based on the significant main effect of sex, males and females were analyzed separately. In females, a one-way ANOVA revealed a significant main effect of group (F(3,22) = 7.22, *p* < 0.001). The MQxFQ-S group demonstrated significantly higher levels of CREB than the MQxFQ-10 and MSxFS-S groups but did not differ from the MQxFQ-10 group. There were no other significant differences between groups. In males, a one-way ANOVA revealed a significant main effect of group (F(3,16) = 9.6, *p* < 0.001). In males, the MQxFQ-S group demonstrated significantly higher CREB levels than all other groups, and there were no other significant differences between groups. This result suggests that CDPPB was not as effective in females as compared to males in reducing CREB in the NAc.

#### 3.8.2. MQxFS Group

CREB is presented as a function of group in Figure 8C. A two-way ANOVA revealed no significant main effect of sex. Therefore, we dropped sex as a factor, and a one-way ANOVA revealed a significant main effect of group (F(3,35) = 10.96, *p* < 0.001). Identical to the MQxFQ group, there was only a significant difference between the MSxFS-S and MQxFS-S groups, and the groups administered either dose of CDPPB did not significantly differ from controls (MSxFS-S) or the MQxFS-S group.

#### 3.8.3. MSxFQ Group

CREB is presented as a function of group in Figure 8D. A two-way ANOVA revealed no significant main effect of sex. Therefore, we dropped sex as a factor, and a one-way ANOVA revealed a significant main effect of group (F(3,35) = 3.94, *p* < 0.017). In this case, the MSxFQ-S group was significantly higher than the MSxFS-S and MSxFQ-30 groups but not the MSxFQ-10 group. Thus, only the 30 mg/kg CDPPB dose was effective in this group in reducing CREB to control levels.

### 3.9. F1-Generation Comparison of Offspring of NQ-Treated Rats in PPI and Acoustic Startle

One final comparison was performed to compare saline-treated MSxFS, MQxFQ, MQxFS, and MSxFQ animals to determine if the MQxFQ group demonstrated more severe deficits in PPI than the other founder crosses. Prepulse inhibition (%) is presented as a function of decibel of prepulse and group for females in Figure 9A and for males in Figure 9C. A three-way ANOVA with sex, group, and decibel of prepulse revealed a significant main effect of sex (F(1,75) = 13.59, *p* < 0.001) and founder cross (F(3,75) = 28.55, *p* < 0.001) and a significant two-way interaction of group x decibel of prepulse (F(6,150) = 2.19, *p* < 0.047). Overall, females demonstrated significantly improved performance as compared to males, as we revealed above. Based on the significant main effect of sex, we analyzed females and males separately. In females, a two-way ANOVA revealed a significant main effect of group (F(3,36) = 10.58, *p* < 0.001) and a significant two-way interaction of group x decibel of prepulse (F(6,72) = 2.90, *p* < 0.014). At 73 and 76 dB, the MQxFQ group was significantly less than all three other groups. However, at 82 dB, the MQxFQ and MSxFQ groups were equivalent but significantly less than the MQxFS group, as well as the MSxFS control group. In males, a three-way ANOVA revealed a significant main effect of group (F(3,39) = 20.25, *p* < 0.001), but the interaction of group x decibel of prepulse was not significant. Across all three decibels of prepulse, the MQxFQ founder cross demonstrated a significant deficit relative to the MQxFS founder cross (and MSxFS controls) but was equivalent to the MSxFQ group. In sum, within the MQxFQ group, females demonstrated a more robust PPI deficit than in males relative to the other crosses. However, in males, the MSxFQ cross was equivalent to MQxFQ in terms of the deficit in PPI.

Results for acoustic startle are presented in Figure 9B for females and Figure 9D for males. There were no group differences in acoustic startle.

### 3.10. F1-Generation Comparison of Offspring of NQ-Treated Rats in NAc CREB

CREB is presented as a function of group in Figure 10. A two-way ANOVA did not reveal any significant main effect of sex, so this was dropped as a factor. The resulting one-way ANOVA revealed a significant main effect of group (F(3,36) = 23.27, *p* < 0.001). The MQxFQ-S group demonstrated a significantly higher level of CREB in the NAc than all other groups, and the MQxFS-S and MSxFQ-S groups also demonstrated a significantly higher level of CREB than MSxFS.

## 4. Discussion

This study demonstrates that stimulation of either the A(2A) or mGlu5 receptor results in an attenuation of sensorimotor gating deficits and a reduction in elevated CREB in the NAc in an established and validated heritable rodent model of psychosis. In experiment 1, administration of CGS21680 alleviated PPI deficits and significant increases in CREB in the NAc, and in experiment 2, administration of CDPPB was also showed efficacy in alleviating sensorimotor gating deficits in this model. This study is the first to show these findings in a heritable model of psychosis, with phenotypes that are consistent with SZ presenting in the offspring of NQ-treated animals. MQxFQ rats demonstrated the most robust deficits in PPI compared to all other founder crosses; aside from the MSxFQ rats, deficits were equivalent. This enhanced deficit is consistent with clinical genetic findings that indicate significantly increased risk of development of psychosis if an individual is the offspring of two affected parents [38]. In addition, the MQxFQ cross also demonstrated the most prominent increase in accumbal CREB, which may also be related to findings from RNAseq analysis in our previous work that reported that this group demonstrated a significant increase in the cortisol synthesis pathway in the dorsal striatum [21]. The findings here generally indicate that both treatment targets, the adenosine A(2A) receptor and the mGlu5 receptor, demonstrate potential for alleviating sensorimotor gating deficits in a model system of enhanced dopamine D2 receptor sensitivity. We previously reported that CDPPB was also effective in alleviating the enhanced behavioral response to nicotine in conditioned place preference expression and reinstatement, indicating that this target may have the potential to be effective in both alleviating sensorimotor gating deficits and dampening the rewarding impact of nicotine.

In experiments 1 and 2, among the MQxFQ group, females performed better in PPI overall compared to males, but administration of either dose of CGS21680 (0.03 mg/kg or 0.09 mg/kg) was successful in alleviating deficits in PPI to levels equal to MSxFS controls. This study is the first to demonstrate a sex difference in PPI within Sprague–Dawley rats. Previously, female Wistar rats have been shown to demonstrate more robust PPI deficits than males [39]; thus, this effect appears to be strain-dependent. In addition, the administration of either dose of CGS21680 reduced enhancements in CREB in MQxFQ rats relative to control animals. This is significant, as D_1_ and D_2_ receptor activation has been associated with CREB activation in the NAc [40]. This may indicate that stimulation of the A(2A) receptor results in a reduction in D_2_ signaling by acting through striatal heteromers [41]. Sustained increases in accumbal CREB have also been associated with negative symptom presentation in psychosis [42], which may be related to the sensorimotor gating deficits observed here.

MQxFS rats demonstrated significant PPI deficits compared to the MSxFS control group, but these deficits were not as severe as deficits observed in MQxFQ rats. In experiment 1, both doses of CGS21680 attenuated PPI deficits in male subjects; however, the lower 0.03 mg/kg dose was not effective in females. This finding was mirrored in the MSxFQ epigenetic cross, where both doses of CGS21680 successfully attenuated deficits in males but no differences were found within drug groups among females. In the clinical population, it has been reported that females present 40% lower A(2A) receptor expression at the mRNA level compared to males [43]. This may explain the reduced behavioral response to CGS21680 observed in females in our MQxFS and MSxFQ epigenetic crosses. In Sprague–Dawley rats, no significant sex differences have been found regarding binding affinity and selectivity of the A(2A) receptor [44]. However, this could indicate that the differential expression of these receptors is primarily responsible for the observed sex differences in PPI and not the affinity of the receptor itself. Interestingly, we found that the startle response was significantly reduced in MQxFS rats by the lower 0.03 mg/kg dose of CGS21680, but no differences in startle were observed among other epigenetic crosses.

Among MQxFS and MSxFQ subjects, 0.03 mg/kg and 0.09 mg/kg CGS21680 attenuated significant increases in CREB observed in saline-treated rats. No sex differences were observed in total CREB in experiment 1, a finding previously shown in Sprague–Dawley rats [45]. These findings suggest a significant role of the A(2A) receptor in psychosis pathology and sensorimotor gating deficits. A knockout of the A(2A) receptor has been previously shown to induce PPI deficits [46]. Stimulation of the A(2A) receptor prior to testing restored deficits in adenosine modulation and sensorimotor gating previously associated with negative and cognitive symptoms of SZ [47]. Additionally, the consistent restoration of CREB levels to those of controls in the NAc across sex and epigenetic crosses suggests that CREB-linked emotional reactivity may be heavily involved in sensorimotor gating responses in psychosis. Future studies will seek to analyze changes in A(2A)-D_2_ heteromers in the striatum in the F1 generation because past work in post-mortem SZ brain tissue has shown reductions in D2-A(2A) heteromers in the caudate nucleus [48]. The CREB response to restraint stress and nicotine reinstatement will additionally be explored to determine the role of CREB in drug-induced plasticity and emotional responses to stress.

In experiment 2, the mGlu5 PAM CDPPB attenuated PPI deficits and normalized enhanced CREB in the NAc but with less overall efficacy than CGS21680. Sex differences were observed within all epigenetic crosses in experiment 2. As in experiment 1, MQxFQ females performed better than males. Interestingly, the 10 mg/kg dose of CDPPB was more effective in female MQxFQs, whereas the 30 mg/kg dose was more effective in males. Clinically, females have demonstrated increased levels of glutamate in the striatum compared to males [49]. It has been shown mGlu5 receptors, in particular, are expressed at higher levels in females [50]. This may indicate that females are more sensitive to glutamate and explain why the lower dose of CDPPB was more effective in PPI. The 30 mg/kg dose of CDPPB may not be as efficacious in females because of the increased number of mGlu5 receptors in females, but this has yet to be determined. In males, as expected, the 10 and 30 mg/kg doses of CDPPB were effective in reducing enhanced levels of CREB in the NAc. Females demonstrated higher levels of total CREB, and CDPPB was not as effective. This enhanced baseline increase in females may be responsible for the lack of effectiveness of CDPPB in reducing CREB to control levels. Female rats have previously demonstrated enhanced excitatory signaling in the NAc compared to males [51], which may be related to this effect. Increased signaling in the D_2_-rich mesolimbic pathway may also play a role in generating increases in CREB, as in vitro quinpirole treatment has been shown to enhance the expression and activation of CREB [52].

A(2A) agonist CGS21680 demonstrated greater efficacy than CDPPB in reducing accumbal CREB across the founder crosses. The precise mechanism of this effect is unknown. However, it is important to note that A(2A) agonist CGS 21680 has a dopamine D2 receptor antagonistic-like action, in that it indirectly decreases dopamine D2 signaling. Past work has shown that dopamine D2 antagonist sulpiride slows increases in phosphorylated CREB produced by D1 agonist SKF 38393 in the caudate nucleus [53]. Interestingly, antipsychotic clozapine, given alone, was shown to decrease phosphorylation of CREB in the mouse dorsal striatum, which may be related to its ability to inhibit ERK1/2 [54] and reduce intracellular calcium levels. Using in vitro preparations, Fuxe and colleagues reported that CGS 21680 was effective at reducing increases in intracellular calcium induced by quinpirole [55]. On the other hand, CDPPB, as a positive allosteric modulator of mGlu5, likely works through a different mechanism. A 10 mg/kg dose of CDPPB has been shown to increase the phosphorylation of CREB in the prefrontal cortex and hippocampus in past work, but a 30 mg/kg dose of CDPPB was not shown to have any effect [37]. This is the first study to analyze the effect of CDPPB on a system in which CREB is elevated, as well as the effects of CDPPB specifically on CREB in the NAc.

The MQxFS epigenetic cross demonstrated similar results to MQxFQ rats in PPI, although the CREB levels in the NAc were reduced compared to the MQxFQ group. Sex differences were noted in both crosses. Whereas the 10 mg/kg dose of CDPPB was generally ineffective across the male and female MQxFS groups, the 30 mg/kg dose was far more effective in male animals. Interestingly, males demonstrated an enhanced startle response that was significantly reduced by the 30 mg/kg dose of CDPPB. Male Long–Evans rats have previously demonstrated enhancements in acoustic startle response [37]. The dose dependency of CDPPB has been previously explored in novel object recognition memory, with lower doses displaying greater efficacy [37]. Our findings demonstrate that the dose dependency of CDPPB is not only affected by the behavioral task but also by sex and founder cross. CREB levels in the NAc of the MQxFS group support our original hypothesis, as both 10 and 30 mg/kg of CDPPB reduced CREB to control levels. Both doses were successful in attenuating aberrant CREB levels, likely due to the D2-genotype/phenotype contributions coming from only one founder. Clinically, having only one parent diagnosed with psychosis, as opposed to two, decreases the chance of illness development by over 20% [56].

MSxFQ rats, likewise, demonstrated sex differences in PPI, with neither dose of CDPPB alleviating PPI deficits observed in saline-treated animals. PPI deficits in MSxFQ males were rescued by the 30 mg/kg dose of CDPPB. Interestingly, MSxFQ males given saline did not demonstrate PPI deficits at 82 dB. Our laboratory has previously hypothesized that the male parent contributes more to the psychosis phenotype via epigenetics, as MSxFQ rats consistently perform better than their MQxFS counterparts. One such instance of increased risk due to paternal genetic contributions arises when the male is significantly older. Specific mutant clone lines expand as men age, which can contribute to changes in neurodevelopment, which increases risk of neuropsychiatric disorders [57]. MSxFQ rats given 30 mg/kg CDPPB demonstrated significantly reduced CREB in the NAc; however, the 10 mg/kg dose failed to lower CREB. In summary, mGlu5 PAM CDPPB demonstrated reduced efficacy compared to CGS21680 across all epigenetic crosses in both PPI and levels of CREB. Adenosine is heavily involved in PPI, primarily acting through A(2A) receptors in the NAc [58], suggesting that CGS21680 may be beneficial in alleviating negative cognitive symptoms of SZ. Our laboratory has previously demonstrated the effectiveness of CDPPB in PPI as well but in F0-generation rats that were neonatally treated with quinpirole [13]. Future studies will aim to evaluate the genetic contribution of the male spermatozoa in our epigenetic model of psychosis. Additionally, behavioral changes and responses to CDPPB will be analyzed in varying estrous cycles to determine if this played a role in the numerous sex differences observed in experiment 1.

While our results suggest a positive outlook on the stimulation of the adenosine and glutamate systems in treating psychosis, clinical shortcomings within these areas have been previously documented [59,60]. However, great interest and success in targeting the mGlu5 and A(2A) receptors persists, as the etiology of psychosis remains poorly understood and current antipsychotics continue to induce intolerable side effects. It has recently been shown that CDPPB is effective in restoring cognitive function in an MK-801 model of SZ via reduction of extracellular glutamate in the medial prefrontal cortex [61]. Additionally, CGS21680 has been shown to alleviate antipsychotic-induced tardive dyskinesia in mice [62]. This finding reveals alternative roles for A(2A) stimulation in the management of antipsychotic side effects. The complexity and vast symptomology of psychosis contribute to the difficulty of engineering an effective treatment. With several neurotransmitters and neuromodulators involved in the presentation of symptoms, it is valuable to continue targeting mGlu5 and A(2A) receptors as we further elucidate aspects of psychosis that can contribute to improved patient outcomes.

In conclusion, this study revealed that CGS21680 and CDPPB are effective in rescuing PPI deficits in a transgenerational model of psychosis. However, CGS21680 demonstrated enhanced efficacy over CDPPB; in many cases, only one dose of the drug was successful in alleviating sensorimotor gating deficits. Both drugs additionally reduced CREB in our model of psychosis, with CGS21680 again demonstrating enhanced effectiveness. Several sex differences across groups arose within experiments 1 and 2, which we hypothesize were due to differences in sensitivity to mGlu5 and A(2A) receptors across males and females. These findings have strong implications for both the adenosine and glutamate systems in the treatment of psychosis and raise interest in further analyzing striatal heteromers and their role in psychosis. In addition, an important finding here is the level of severity, which differs across these founder crosses, with the most severe deficit in PPI and the most prominent effect on CREB being present in the MQxFQ cross. This is consistent with clinical work, in that levels of severity do appear to vary across individuals diagnosed with psychosis [63,64], providing further validity for this model. Overall, future work will focus on further modulating these systems to elucidate precise neurocircuitry; reduce supersensitized D_2_ signaling; and, in turn, improve health outcomes for this vulnerable population.

## Figures and Tables

**Figure 1 cells-14-00182-f001:**
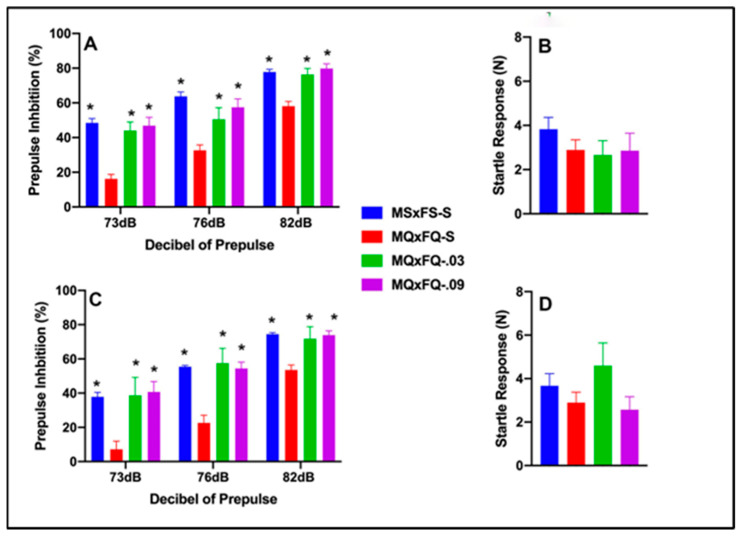
Prepulse inhibition (%) is presented as a function of MQxFQ groups in (**A**) (females) and (**C**) (males). In females and males, the MSxFS, MQxFQ-0.03, and MQxFQ-0.09 groups were equivalent and all significantly greater in PPI than the MQxFQ-S group (indicated by *, *p* < 0.05). Acoustic startle is presented as a function of MQxFQ groups in (**B**) (females) and (**D**) (males). There were no significant differences between groups.

**Figure 2 cells-14-00182-f002:**
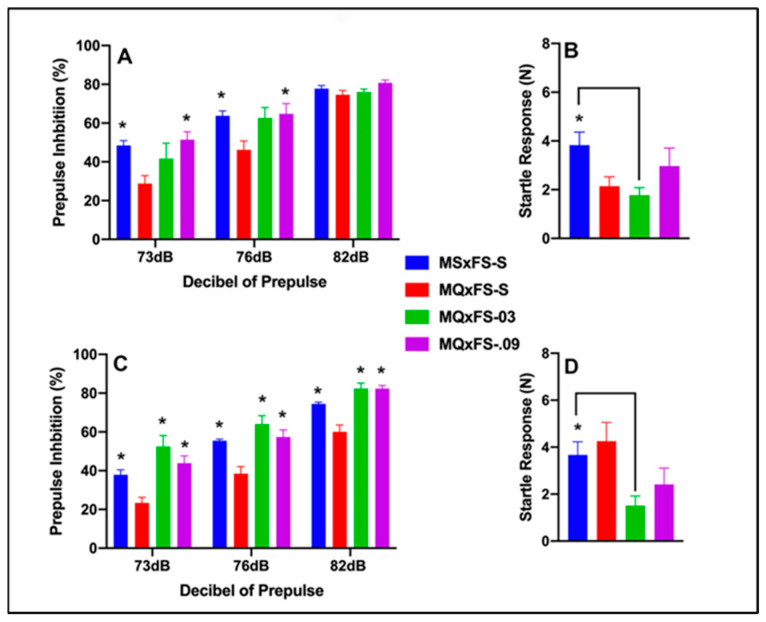
Prepulse inhibition (%) is presented as a function of MQxFS groups in (**A**) (females) and (**C**) (males). In females, at 73 and 76 dB, the MSxFS-S and MQxFS-0.09 groups were equivalent and both significantly greater in PPI than the MQxFS-S group (indicated by *, *p* < 0.05). There were no effects at 82 dB. In males, the MSxFS, MQxFS-0.03, and MQxFS-0.09 groups were equivalent and all significantly greater in PPI than the MQxFS-S group across all decibels of prepulse (indicated by *, *p* < 0.05). Acoustic startle is presented as a function of MQxFQ groups in (**B**) (females) and (**D**) (males). The MSxFS-S group was significantly greater than the MQxFS-0.03 group in both females and males.

**Figure 3 cells-14-00182-f003:**
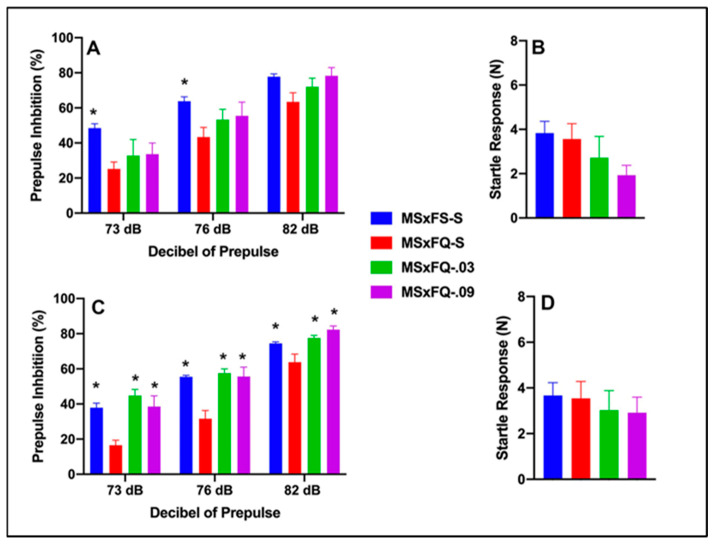
Prepulse inhibition (%) is presented as a function of MSxFQ groups in (**A**) (females) and (**C**) (males). In females, at 73 dB and 76 dB of prepulse, the MSxFS was greater than MSxFQ-S (indicated by *, *p* < 0.05); however, there were no differences between groups given either 0.03 or 0.09 mg/kg CGS 21680 and MSxFS or MSxFQ-S at this level of prepulse. At 82 dB, there were no significant group differences. In males, the MSxFS, MSxFQ-0.03, and MSxFQ-0.09 groups were equivalent and all significantly greater in PPI than the MSxFQ-S group across all decibels of prepulse (indicated by *, *p* < 0.05). Acoustic startle is presented as a function of MQxFQ groups in (**B**) (females) and (**D**) (males). There were no significant differences between groups.

**Figure 4 cells-14-00182-f004:**
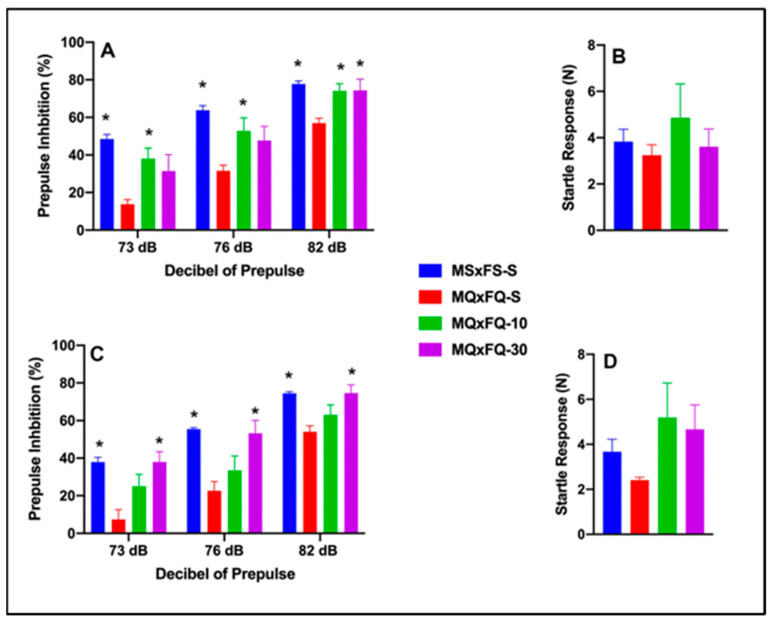
Prepulse inhibition (%) is presented as a function of MQxFQ groups in (**A**) (females) and (**C**) (males). In females, at 73 and 76 dB of prepulse, the MSxFS-S and MQxFQ-10 groups were equivalent and significantly greater than the MQxFQ-S group (indicated by *, *p* < 0.05). At 82 dB, the MSxFS, MQxFQ-10, and MQxFQ-30 groups were all equivalent and significantly greater than the MQxFQ-S group (indicated by *, *p* < 0.05). In males, the MSxFS-S and MQxFQ-30 groups were equivalent and demonstrated significantly greater levels of PPI than the MQxFQ-10 and MQxFQ-S groups at all decibel levels. Acoustic startle is presented as a function of MQxFQ groups in (**B**) (females) and (**D**) (males). There were no significant differences between groups.

**Figure 5 cells-14-00182-f005:**
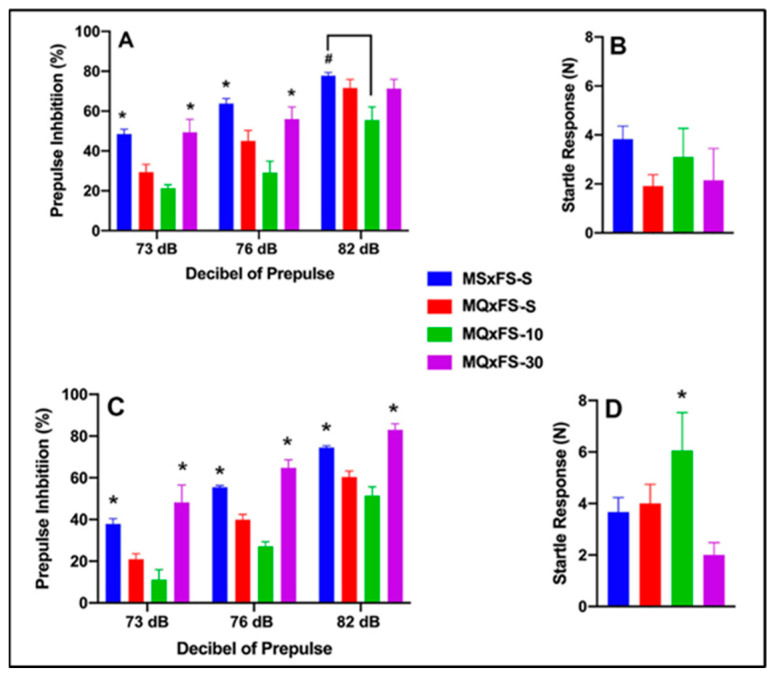
Prepulse inhibition (%) is presented as a function of MQxFS groups in (**A**) (females) and (**C**) (males). In females, the MSxFS-S and MQxFS-30 groups were equivalent at 73 and 76 dB of prepulse, and both demonstrated significantly greater PPI than the MQxFS-S and MQxFS-10 groups (indicated by *, *p* < 0.05). At 82 dB, only the MSxFS-S and MQxFS-10 groups were significantly different (indicated by #, *p* < 0.05). In males, the MSxFS-S and MQxFS-30 groups were equivalent at all dB levels of prepulse, and both demonstrated significantly greater PPI than the MQxFS-S and MQxFS-10 groups across all three dB levels of prepulse (indicated by *, *p* < 0.05). Acoustic startle is presented as a function of MQxFS groups in (**B**) (females) and (**D**) (males). There were no significant group differences in females, but in males, the MQxFS-10 group demonstrated a significant increase in acoustic startle relative to all other groups (indicated by *, *p* < 0.05).

**Figure 6 cells-14-00182-f006:**
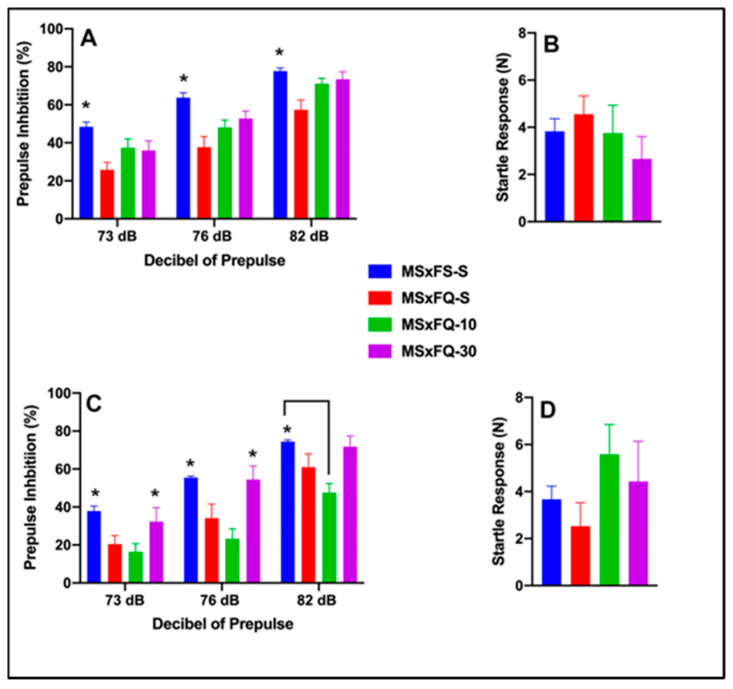
Prepulse inhibition (%) is presented as a function of MSxFQ groups in (**A**) (females) and (**C**) (males). In females, the MSxFS group was equivalent to both the MSxFQ-10 and MSxFQ-30 groups, and MSxFS was significantly greater than the MSxFQ-S group across all dB levels of prepulse (indicated by *, *p* < 0.05). In males, at 73 and 76 dB of prepulse, the MSxFS-S and MSxFS-30 groups were equivalent and significantly greater than the MSxFQ-S group in PPI (indicated by *, *p* < 0.05). At 82 dB of prepulse, only the MSxFS-S group was significantly greater than the MSxFQ-10 group (indicated by * and line, *p* < 0.05). Acoustic startle is presented as a function of MQxFQ groups in (**B**) (females) and (**D**) (males). There were no significant differences between groups.

**Figure 7 cells-14-00182-f007:**
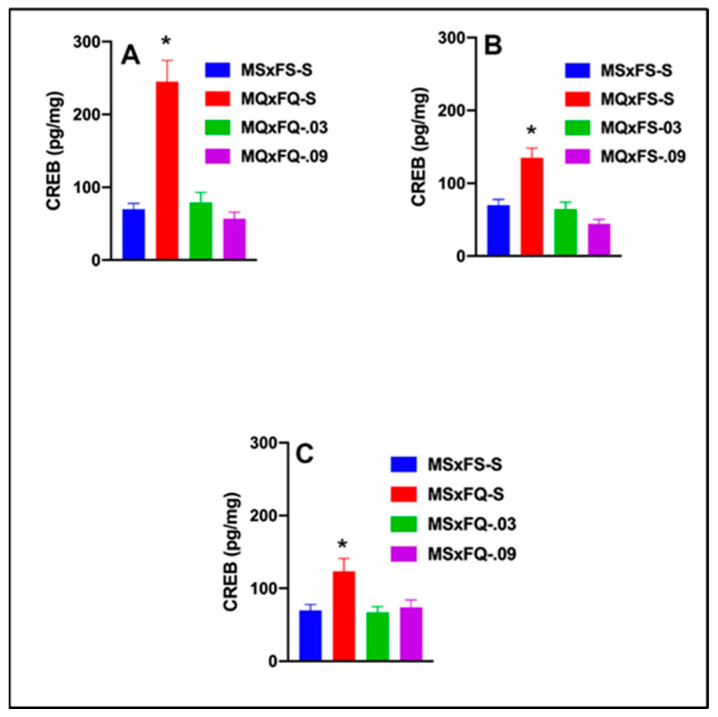
CREB is presented as a function of group for (**A**) MQxFQ, (**B**) MQxFS, and (**C**) MSxFQ in experiment 1. Across all three groups, the MQxFQ-S, MQxFS-S, and MSxFQ-S groups demonstrated significantly greater levels of CREB protein in the NAc than all other groups (indicated by *, *p* < 0.05).

**Figure 8 cells-14-00182-f008:**
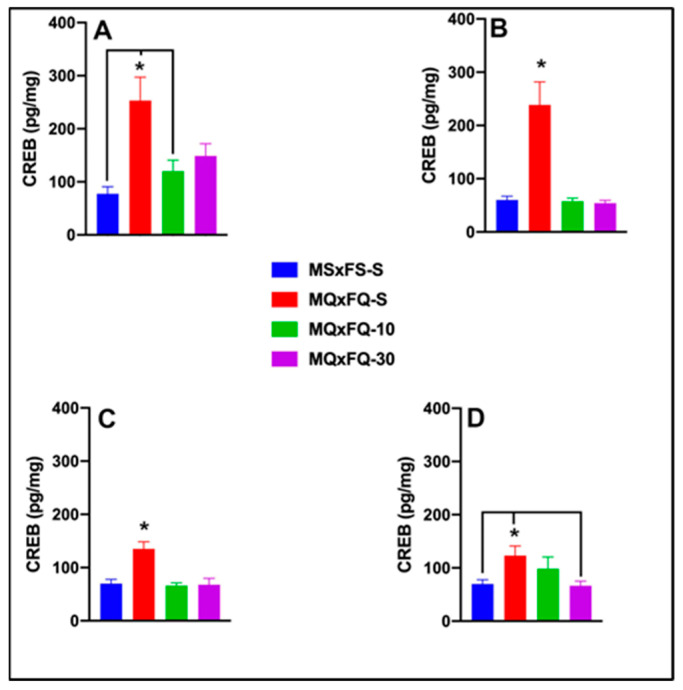
CREB is presented as a function of MQxFQ groups for females (**A**) and males (**B**). CREB is presented as a function of MQxFS groups in (**C**) and MSxFQ groups in (**D**). In female MQxFQ groups, the MQxFQ-S group demonstrated significantly greater CREB protein in the NAc than the MSxFS-S and MQxFQ-10 groups (indicated by *, *p* < 0.05 and line) but not significantly greater levels than the MQxFQ-30 group. In males, the MQxFQ-S group was significantly greater than all other groups (indicated by *, *p* < 0.05). As shown in (**C**), the MQxFS-S group demonstrated significantly greater levels of CREB than all other groups (indicated by *, *p* < 0.05). As shown in (**D**), MSxFQ-S was significantly greater than the MSxFS-S group and MSxFQ-10 group (indicated by * *p* < 0.05 and line) but was not significantly different from the MSxFQ-30 group.

**Figure 9 cells-14-00182-f009:**
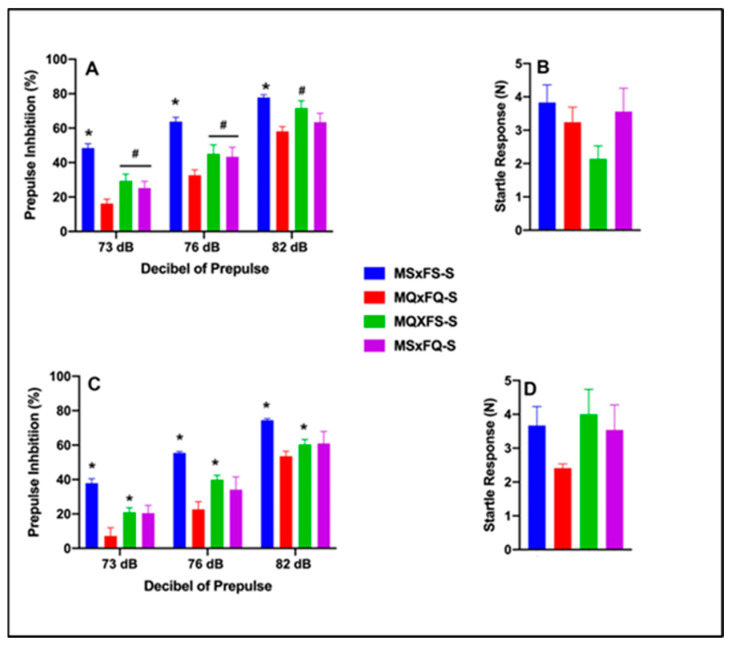
Prepulse inhibition (%) is presented as a function of all saline-treated founder crosses in females (**A**) and males (**C**). In females, the MSxFS-S group was significantly greater than all other groups across all dB values of prepulse (indicated by *, *p* < 0.05). The MQxFS-S group and MSxFQ-S group were significantly greater than the MQxFQ-S group at 73 and 76 dB of prepulse (indicated by #, *p* < 0.05). At 82 dB of prepulse, the MSxFS-S and MQxFS-S groups did not significantly differ but demonstrated greater levels of PPI than the MQxFQ-S and MSxFQ-S groups (indicated by *, *p* < 0.05). In males, the MSxFS-S group demonstrated significantly greater levels of PPI than all other groups across all dB values of prepulse (indicated by *, *p* < 0.05). Across all dB levels, the MQxFS-S group demonstrated significantly improved PPI relative to the MQxFQ-S group (indicated by *, *p* < 0.05). Acoustic startle is presented for females (**B**) and males (**D**). There were no significant differences between groups.

**Figure 10 cells-14-00182-f010:**
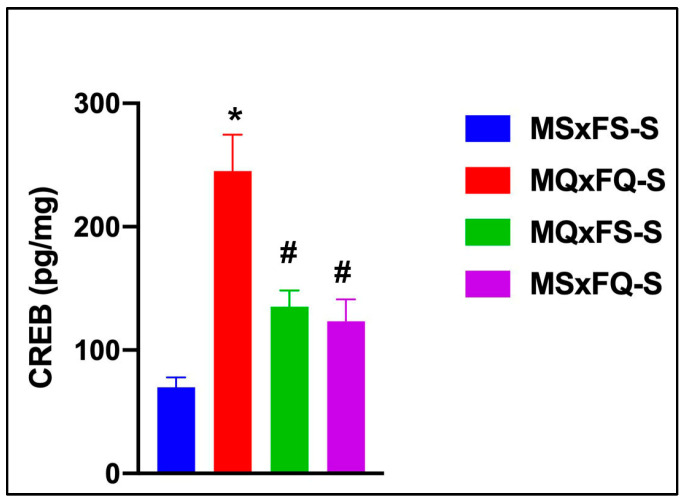
CREB is presented as a function of group for F1-generation vehicle-treated founder crosses. The MQxFQ-S group was significantly higher than all other groups (* indicates *p* < 0.05), and the MQxFS-S and MSxFQ-S groups were significantly higher than the MSxFS-S group (# indicates *p* < 0.05). This result supports the comparison of PPI across founder crosses, in that the MQxFQ-S group demonstrated the highest level of CREB protein in the NAc, as well as the most prominent deficits in PPI across the different founder crosses.

## Data Availability

The data that support the findings of the current study are available from the corresponding author upon reasonable request.

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
