# Peer review of "Investigation of the Roles of the Adenosine A(2A) and Metabotropic Glutamate Receptor Type 5 (mGlu5) Receptors in Prepulse Inhibition and CREB Signaling in a Heritable Rodent Model of Psychosis"

_cells, 2025, doi:10.3390/cells14030182_

Round 1
Reviewer 1 Report
Comments and Suggestions for Authors
Reviewer report on manuscript entitled „Investigation of the roles of the adenosine A(2A) and metabotropic receptor type 5 (mGlu5) receptors in Prepulse Inhibition and CREB Signaling in a Heritable Rodent Model of Psychosis” ( Manuscript ID: cells-3422263)
The present manuscript shows that in a rodent neonatal quinpirole model of psychosis the prepulse inhibition (PPI) deficits were alleviated by both mGlu5 receptor PAM- CDPPB and adenosine A2 receptor agonist CGS21680. Additionally, in this model of psychosis elevated CREB levels in the nucleus accumbens were reduced to the control levels both by CDPPB and CGS21680.
These findings are original, novel and may have implications towards the role of both the adenosine and glutamate systems in the treatment of psychosis
The manuscript is well written, the finding presented here are original and shown for the first time. I have only minor comments:
Figure and the description of the results it represents could be placed on the same page for the clarity of reading
The title line 2 the word metabotropic is missing
Line 763 ref 52 not 53 should be quoted
Line 806 CGS21680 demonstrated enhanced efficacy “of” CDPPB. put “over” CDPPB
Author Response
Comments 1: Figure and the description of the results it represents could be placed on the same page for the clarity of reading.
Response 1: Figure results and captions have been organized such that all information pertaining to individual figures are on the same page.
Comments 2: The title line 2 the word metabotropic is missing.
Response 2: Title spacing has been changed such that all words fully fit on the given lines.
Comments 3: Line 763 ref 52 not 53 should be quoted.
Response 3: References were misnumbered in the bibliography, an extra citation was present that should have been removed in editing. Additional references have been added upon revision. The Fuxe reference is now 55 and all references are accurate.
Comments 4: Line 806 CGS21680 demonstrated enhanced efficacy “of” CDPPB. put “over” CDPPB
Response 4: The correction from "of" to "over" has been made.
Reviewer 2 Report
Comments and Suggestions for Authors
The manuscript (cells-3422263) examined the effects of the mGlu5 PAM CDPPB and the A2a agonist CGS21680 on prepulse inhibition (PPI) tests and levels of CREB in a transgenerational neonatal quinpirole model of psychosis in rats. Both CDPPB and CGS21680 were found to alleviate PPI deficits and increased accumbens CREB levels in the F1 offspring, which were dependent on parental founder cross conditions and sex, and with CGS21680 more effective than CDPPB. It was concluded that targeting the adenosinergic and glutamatergic systems could be of relevance in treatment of psychosis.
Overall, the transgenerational rat model is very interesting and the study was well executed and manuscript well written. My only suggestion to the authors is to tone down a bit on the significance and be more balanced in discussion of the findings, given many failures in clinical translation of preclinical findings on the A2a and mGluR5 systems.
Author Response
Comments 1: Overall, the transgenerational rat model is very interesting and the study was well executed and manuscript well written. My only suggestion to the authors is to tone down a bit on the significance and be more balanced in discussion of the findings, given many failures in clinical translation of preclinical findings on the A2a and mGluR5 systems.
Response 1: Thank you, we agree with this point and have subsequently added the following information to the manuscript on lines 832-845: "While our results suggest a positive outlook on the stimulation of the adenosine and glutamate systems in treating psychosis, clinical shortcomings within these areas have been previously documented [59,60]. However, great interest and success in targeting the mGlu5 and A(2A) receptors persists, as the etiology of psychosis remains poorly understood, and current antipsychotics continue to induce intolerable side effects. It has recently been shown that CDPPB is effective in restoring cognitive function in an MK-801 model of SZ via reduction of extracellular glutamate in the medial prefrontal cortex [61]. Additionally, CGS21680 has been shown to alleviate antipsychotic induced tardive dyskinesia in mice [62]. This finding reveals alternative roles for A(2A) stimulation in the management of antipsychotic side effects. The complexity and vast symptomology of psychosis contributes to the difficulty of engineering an effective treatment. With several neurotransmitters and neuromodulators involved in the presentation of symptoms, it is valuable to continue targeting mGlu5 and A(2A) receptors as we further elucidate aspects of psychosis that can contribute to improved patient outcomes."
Reviewer 3 Report
Comments and Suggestions for Authors
The manuscript studies an interesting topic. The authors use a previously validated heritable rodent model of psychosis. The experiments are not clearly explained. They used an important number of animals, but the number of animals used for different groups is not often the same! Besides, they used female animals that may be involved in the menstrual processes. This may surely modify their hormonal assets. To avoid this problem, the number of female animals should be higher than the number of male animals!
In the result section, the authors didn't cite the number of figures they analyzed, which makes it difficult to follow the results! Besides, in some sections, the number of animals wasn't quoted.
The authors should explain how they have decided to use those quantities of quinpirole HCl and adenosine A(2A) agonist, CGS 21680.
In my opinion, this study is interesting, but it is not designed appropriately. The authors should significantly improve the manuscript, descriving better the results.
Strangely, the abstract seems quite clear but the manuscript no.
Author Response
Response to Reviewer 3:
We agree with the reviewer that the stage of the estrous cycle in females could possibly affect the results observed here. However, to fully analyze all of the stages and influence of the estrous cycle would require a likely 3-4 fold increase in number of females. Further, this was not a focus of this project. Adding another variable would also complicate the research design, and there would be 4 factors in the design if this was included. Thus, with four factors, a significant four-way interaction cannot be interpreted. We have approached this on p. 4, lines 168-171.
The numbers of animals in each experiment has now been described throughout the results, and the number of figures has been checked.
The doses of quinpirole, CGS 21680 and CDPPB were all based on our past work and the literature. This is now described on page 3, lines 143-146 for CGS 21680, and page 5, 234-235.
We have clarified the research design in the Research Design section on page 4, lines 161-167.
We are unsure about where the manuscript is unclear, although on a thorough revision we did find some wording issues that may have been interpreted as confusing. We have gone through the manuscript and clarified parts of the manuscript that may be confusing.
Round 2
Reviewer 3 Report
Comments and Suggestions for Authors
The authors have addressed my concerns. Therefore, I believe the manuscript may be accepted.